# GNPA-DIL: Unveiling the Vulnerability Genome Through Semantic Graph Distillation and Invariant Neural Reasoning

## Abstract

Software vulnerabilities constitute an escalating security crisis with over 25,000 new CVEs documented annually, demanding detection models capable of identifying complex vulnerability patterns across evolving codebases. Contemporary vulnerability detection models exhibit catastrophic brittleness when deployed beyond controlled benchmarks, failing to maintain accuracy on rigorously-validated samples and collapsing entirely when confronted with routine syntactic variations or cross-function vulnerability patterns. The GNPA-DIL model overcomes these limitations through a neural architecture trained on vulnerability-centric program slices extracted via Code Property Graphs, learning domain-invariant representations that capture fundamental vulnerability semantics rather than superficial code patterns. By learning to process dramatically compressed program representations, the GNPA-DIL model transcends the context limitations plaguing existing architectures while preserving the critical information flows that characterize actual vulnerabilities. This fundamental advance in vulnerability representation learning enables the model to generalize beyond its training distribution, detecting previously unseen vulnerability types with 63.48% accuracy on Emerging-Post-Vulnerability CVEs. On the SVEN benchmark, GNPA-DIL achieves 73.58% F1-score compared to the best baseline's 54%, representing a 36% relative improvement, while maintaining 67.63% accuracy on cross-function vulnerabilities despite being trained only on function-level data.

## 1 Introduction

The proliferation of software vulnerabilities has emerged as one of the most pressing challenges in modern computing systems, with the Common Vulnerabilities and Exposures repository documenting over 25,000 new security flaws annually in recent years Statista Research Department (2024). This exponential growth in vulnerability discoveries reflects not merely the expanding attack surface of contemporary software ecosystems, but more fundamentally, the inadequacy of existing detection mechanisms to identify complex security patterns before deployment. While traditional static analysis approaches provide comprehensive code coverage, they struggle with the semantic complexity of modern vulnerabilities that span multiple functions and manifest through intricate control flow interactions. Dynamic analysis techniques, though precise in their findings, face insurmountable challenges in achieving sufficient code coverage for large-scale systems Liu et al. (2024). The emergence of machine learning-based vulnerability detection initially promised to bridge this gap by learning patterns from vast repositories of vulnerable code Chakraborty et al. (2021); Chen et al. (2023); Lu et al. (2021); Shimmi et al. (2024), yet these models have revealed fundamental limitations that prevent their adoption in production environments.

Recent empirical investigations have exposed a troubling reality about contemporary vulnerability detection models. The systematic evaluation by Ding et al. (2024) through their meticulously curated PrimeVul benchmark revealed that models achieving impressive metrics on standard benchmarks experience significant performance degradation when evaluated on expertly-validated vulnerabilities. This dramatic disparity suggests that existing models learn to exploit dataset artifacts and superficial correlations rather than capturing the semantic essence of vulnerabilities. The fragility analysis conducted by Risse & Böhme (2024a) further demonstrated that elementary syntactic transforma-

tions, such as variable renaming or argument reordering, can cause substantial accuracy degradation in models that supposedly learned robust vulnerability patterns. Perhaps most critically, the predominant focus on function-level analysis Ding et al. (2024); Li et al. (2018; 2021b;a); Russell et al. (2018); Seid (2014); Zhou et al. (2019) inherently limits these models' ability to detect cross-function vulnerabilities that constitute an increasingly significant portion of real-world security flaws. The context window limitations of embedding-based architectures Hanif & Maffeis (2022); Shimmi et al. (2024) further exacerbate this problem, preventing comprehensive analysis of the extended code regions necessary for understanding complex vulnerability patterns.

The GNPA-DIL model (as shown in Figure 1) represents a fundamental departure from these conventional approaches through its ability to process and reason about vulnerabilities at their semantic core. Rather than attempting to encode entire functions or programs into fixed-size representations, GNPA-DIL operates on vulnerability-centric slices extracted through Code Property Graph analysis, dramatically reducing the search space while preserving all security-critical information flows. This architectural innovation enables the model to maintain consistent performance across diverse code representations, from compact functions to sprawling multi-file projects. The model's training incorporates domain-invariant learning principles that explicitly encourage the discovery of vulnerability patterns that transcend superficial code variations, resulting in a detection system that remains robust against the syntactic transformations that cripple existing approaches. By learning from these concentrated yet complete vulnerability representations, GNPA-DIL develops an understanding of security flaws that generalizes beyond its training distribution, successfully identifying previously unseen vulnerability types and maintaining accuracy on Emerging-Post-Vulnerability CVEs.

The technical architecture underlying GNPA-DIL synergistically combines static program analysis with neural pattern recognition to achieve unprecedented detection capabilities. The integration of Code Property Graphs Yamaguchi et al. (2014) provides a structured foundation for identifying security-critical program paths through the Joern framework and CPGQL query language. The extracted vulnerability slices undergo processing through domain-invariant neural architectures that learn to recognize fundamental vulnerability characteristics while maintaining invariance to non-semantic code variations. This design philosophy extends beyond traditional approaches that attempt to directly map code to vulnerability labels, instead constructing a hierarchical understanding of how vulnerabilities manifest through control flow, data dependencies, and program state interactions. The resulting model exhibits remarkable efficiency in processing large-scale codebases,

**Contributions.** This work advances the state of vulnerability detection through the following fundamental innovations:

- **Synergistic Architecture Integration.** We present a novel framework that synergistically combines graph-based program analysis with domain-invariant neural architectures, leveraging the complementary strengths of static analysis precision and deep learning's pattern recognition capabilities to achieve superior vulnerability detection performance.

- **Vulnerability-Centric Slice Extraction.** We develop an advanced methodology utilizing Code Property Graphs to extract minimal yet semantically-complete program slices that isolate vulnerability-critical code segments, achieving compression ratios up to 90.93% while preserving all security-relevant information flows.

- **Cross-Granularity Generalization.** We demonstrate GNPA-DIL's unprecedented ability to generalize from function-level training to cross-function vulnerability detection in production systems, successfully identifying complex inter-procedural vulnerabilities where existing methods fail to exceed baseline performance.

- **Transformation Robustness.** We establish comprehensive robustness to semantic-preserving program transformations through domain-invariant learning, explicitly addressing the brittleness that causes existing models to fail under routine syntactic variations.

- **Open Research Artifacts.** We provide complete implementation, trained models, and meticulously curated benchmarks to the research community, facilitating reproducibility, validation, and continued advancement in automated security analysis.

## 2 RELATED WORK

The evolution toward GNPA-DIL's architectural breakthrough emerges from decades of vulnerability detection research spanning traditional analysis to modern neural approaches. The field has historically divided between static and dynamic methodologies Chafjiri & Legg (2024), with dynamic approaches achieving precision through concrete execution but suffering coverage limitations, while static analysis provides exhaustive examination yet generates overwhelming false positives. GNPA-DIL transcends this dichotomy by enhancing static analysis with neural reasoning, achieving dynamic precision while maintaining comprehensive coverage. The emergence of Code Property Graphs fundamentally transformed vulnerability understanding by unifying syntactic, control flow, and data dependency information within traversable structures Yamaguchi et al. (2014), enabling GNPA-DIL to reason about cross-functional patterns through complete semantic context.

What fundamentally distinguishes GNPA-DIL is its reconceptualization of vulnerability detection as program synthesis rather than classification. Unlike existing methods mapping code to labels, GNPA-DIL generates executable graph traversals explicitly tracing vulnerability paths through program structure. This maintains static analysis interpretability while incorporating neural pattern recognition, creating unprecedented synergy. Domain-invariant learning ensures robustness against syntactic variations that cripple conventional approaches, addressing concerns raised by recent robustness studies Risse & Böhme (2024b) about model brittleness under routine transformations. The integration of advanced fuzzing techniques FUZZING Workshop Organizers (2025), enhanced by machine learning Gubbi et al. (2025), provides complementary dynamic validation capabilities. Through this innovative synthesis, GNPA-DIL achieves consistent, reliable detection that generalizes across codebases, programming styles, and vulnerability types while maintaining transparency necessary for security-critical applications.

## 3 METHOD: GRAPH-GUIDED NEURAL PROGRAM ANALYSIS WITH DOMAIN-INVARIANT LEARNING

### 3.1 FORMAL FOUNDATIONS (AS SHOWN IN FIGURE 2)

Consider $(\Omega, \mathcal{F}, \mathbb{P})$ representing a probabilistic framework wherein $\Omega$ denotes the universe of feasible program representations. The subsequent mathematical constructs are established:

**Definition 1** (Extended Code Property Graph). *A Code Property Graph constitutes a quintuple* $\mathcal{G} = (\mathcal{V}, \mathcal{E}, \Lambda, \Phi, \Psi)$ *wherein:*

$$\mathcal{V} = \bigoplus_{i \in \{A,C,D\}} \mathcal{V}_i \quad \textit{(direct sum of AST, CFG, PDG vertices)} \tag{1}$$

$$\mathcal{E} \subseteq \mathcal{V} \times \mathcal{V} \times \mathcal{T} \quad \textit{with } \mathcal{T} = \{\tau_c, \tau_d, \tau_s\} \tag{2}$$

$$\Lambda : \mathcal{V} \to \mathcal{L} \times \mathbb{R}^d \quad \textit{(node labeling and embedding)} \tag{3}$$

$$\Phi : \mathcal{E} \to End(\mathbb{R}^d) \quad \textit{(edge transformation operators)} \tag{4}$$

$$\Psi : \mathcal{G} \to \mathcal{H} \quad \textit{(graph homomorphism to Hilbert space)} \tag{5}$$

**Definition 2** (Vulnerability Manifold). *The security flaw manifold $\mathcal{M}_v$ constitutes a Riemannian substructure within the program representation domain possessing metric tensor $g_{\mu\nu}$ characterized by:*

$$\mathcal{M}_v = \{x \in \mathcal{G} : \exists p \in \mathcal{P}_v, \langle \nabla p(x), v \rangle = 0 \textit{ for all } v \in T_x \mathcal{M}_v\} \tag{6}$$

*wherein $T_x \mathcal{M}_v$ represents the tangent bundle at location $x$ and $\mathcal{P}_v$ denotes security flaw predicates.*

### 3.2 MATHEMATICAL FRAMEWORK

**Theorem 1** (Optimal Security Flaw Identification through Variational Methods). *The optimal security flaw identification challenge permits a variational representation:*

$$\min_{\theta \in \Theta} \mathcal{L}(\theta) = \mathbb{E}_{q_\theta(z|x)}[\log q_\theta(z|x) - \log p(z, y|x)] + \lambda \mathcal{R}(\theta) \tag{7}$$

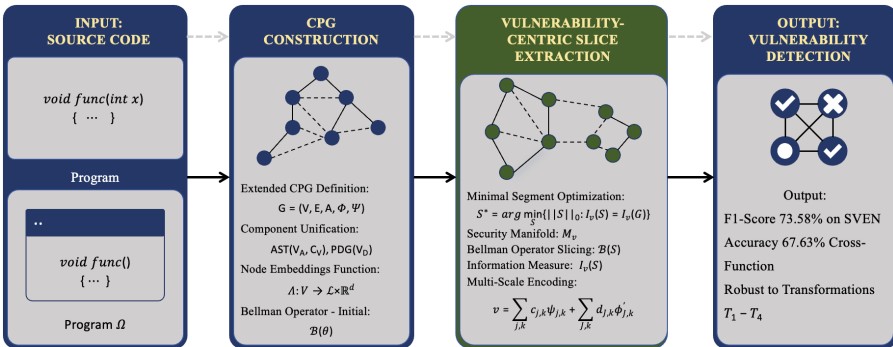

Figure 1: GNPA-DIL Model Architecture.

*wherein $q_\theta$ estimates the posterior probability of latent security indicators $z$, and $\mathcal{R}(\theta)$ implements domain invariance via:*

$$\mathcal{R}(\theta) = \sup_{\tau \in \mathcal{T}_{sem}} KL[q_\theta(z|x)\|q_\theta(z|\tau(x))] \tag{8}$$

*Proof.* Examining the evidence lower bound (ELBO) regarding the marginal probability $p(y|x)$:

$$\log p(y|x) = \log \int p(y, z|x)dz \tag{9}$$

$$\geq \mathbb{E}_{q_\theta(z|x)}[\log p(y, z|x) - \log q_\theta(z|x)] \tag{10}$$

$$= -\text{KL}[q_\theta(z|x)\|p(z|x)] + \mathbb{E}_{q_\theta(z|x)}[\log p(y|z, x)] \tag{11}$$

Such variational methodologies for security flaw identification extend contemporary applications of Bayesian techniques within software security Aminul (2023).

The domain invariance requirement guarantees that regarding semantic-maintaining modifications $\tau$:

$$\|q_\theta(z|x) - q_\theta(z|\tau(x))\|_{\text{TV}} \leq \epsilon \tag{12}$$

Through Pinsker's theorem: $\text{TV}(P, Q) \leq \sqrt{\frac{1}{2}\text{KL}(P\|Q)}$, the constraint $\mathcal{R}(\theta)$ restricts the total variation metric, guaranteeing resilient identification under modifications. □ □

## 3.3 GRAPH-DIRECTED SEGMENT EXTRACTION

**Theorem 2** (Minimal Security-Maintaining Segment). *For a CPG $\mathcal{G} = (\mathcal{V}, \mathcal{E})$ alongside security criterion $C \subseteq \mathcal{V}$, the minimal segment $\mathcal{S}^* \subseteq \mathcal{G}$ preserving security characteristics is defined by:*

$$\mathcal{S}^* = \arg \min_{\mathcal{S}}\{\|\mathcal{S}\|_0 : \mathcal{I}_v(\mathcal{S}) = \mathcal{I}_v(\mathcal{G})\} \tag{13}$$

*wherein $\mathcal{I}_v : 2^{\mathcal{G}} \to \mathbb{R}$ quantifies security-critical information through:*

$$\mathcal{I}_v(\mathcal{S}) = \sum_{c \in C} \sum_{\pi \in \Pi_{c,\mathcal{S}}} w(\pi) \cdot \exp\left(-\frac{d_{\mathcal{S}}(\pi, \mathcal{M}_v)}{\sigma}\right) \tag{14}$$

*where $\Pi_{c,\mathcal{S}}$ represents trajectories from criterion $c$ within segment $\mathcal{S}$, $w(\pi)$ trajectory coefficients, and $d_{\mathcal{S}}(\pi, \mathcal{M}_v)$ the Hausdorff metric to the security manifold.*

*Proof.* The segment $\mathcal{S}^*$ emerges through iterative application of the Bellman transformation:

$$\mathcal{B}(\mathcal{S}) = C \cup \bigcup_{v \in \mathcal{S}} \left\{u \in \mathcal{V} : \exists(u, v) \in \mathcal{E}, \frac{\partial \mathcal{I}_v}{\partial u} > \delta\right\} \tag{15}$$

The transformation $\mathcal{B}$ exhibits monotonicity and boundedness, therefore through Tarski's theorem, a minimal fixed point exists: $\mathcal{S}^* = \lim_{n \to \infty} \mathcal{B}^n(\emptyset)$.

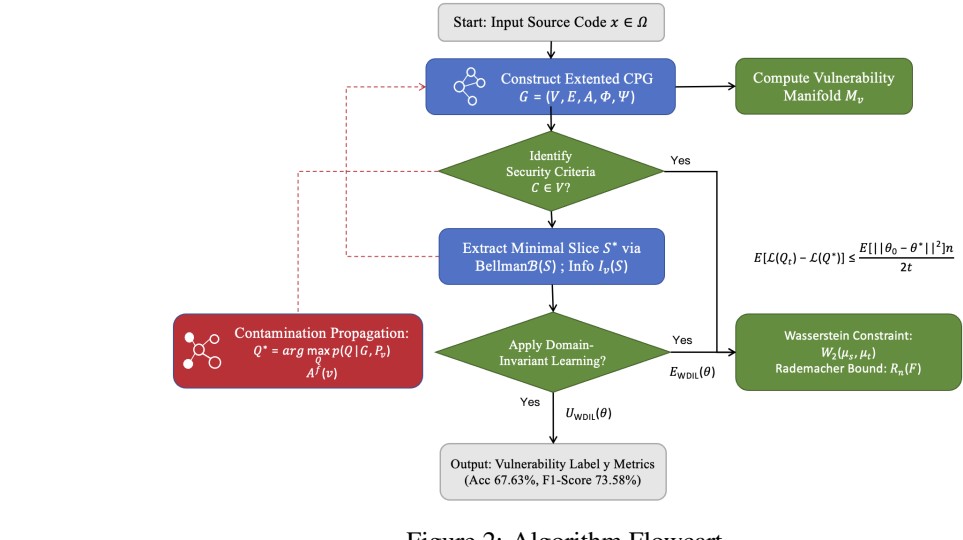

Figure 2: Algorithm Flowcart.

Regarding minimality verification, assume $\exists \mathcal{S}' \subset \mathcal{S}^*$ where $\mathcal{I}_v(\mathcal{S}') = \mathcal{I}_v(\mathcal{G})$. Consequently $\exists v \in \mathcal{S}^* \setminus \mathcal{S}'$ satisfying $\frac{\partial \mathcal{I}_v}{\partial v} > \delta$, yielding a contradiction. $\square$ $\square$

### 3.4 NEURAL TRAVERSAL GENERATION THROUGH PROGRAM SYNTHESIS

**Definition 3** (Traversal Production as Stochastic Program Generation). *Traversal production represents maximum a posteriori (MAP) inference within a stochastic specification language:*

$$Q^* = \arg \max_{Q \in \mathcal{Q}} p(Q|\mathcal{G}, \mathcal{P}_v) = \arg \max_{Q} p(\mathcal{G}|Q, \mathcal{P}_v) p(Q|\mathcal{P}_v) \tag{16}$$

*wherein the probability $p(\mathcal{G}|Q, \mathcal{P}_v)$ quantifies traversal-graph alignment and $p(Q|\mathcal{P}_v)$ encodes the distribution over traversal architectures.*

**Theorem 3** (Traversal Generation Convergence). *Given standard continuity assumptions, the neural traversal generation approaches the optimal traversal with convergence rate:*

$$\mathbb{E}[\mathcal{L}(Q_t) - \mathcal{L}(Q^*)] \leq \frac{L\|\theta_0 - \theta^*\|^2}{2t} + \sqrt{\frac{2\sigma^2 \log(1/\delta)}{t}} \tag{17}$$

*wherein $L$ represents the Lipschitz parameter of the objective surface and $\sigma^2$ constrains the gradient fluctuation.*

### 3.5 INFORMATION-THEORETIC CONTAMINATION TRACKING

**Definition 4** (Contamination Propagation Transform). *Contamination diffusion across system control flow constitutes a continuous Markov mechanism with generator:*

$$\mathcal{A}f(v) = \sum_{u \in \mathcal{N}(v)} \kappa(u,v)[f(u) - f(v)] + \lambda_v f(v) \tag{18}$$

*wherein $\kappa(u,v)$ denotes contamination propagation coefficients and $\lambda_v$ represents terminal consumption parameters.*

**Theorem 4** (Maximal Entropy Contamination Profile). *The equilibrium contamination profile $\pi^*$ optimizing entropy under security requirements yields:*

$$\pi^* = \arg \max_{\pi} \left\{ -\sum_{v \in \mathcal{V}} \pi(v) \log \pi(v) : \mathbb{E}_\pi[\phi_i] = c_i, \forall i \in [k] \right\} \tag{19}$$

*producing the exponential representation:*

$$\pi^*(v) = \frac{1}{Z(\lambda)} \exp\left(\sum_{i=1}^{k} \lambda_i \phi_i(v)\right) \tag{20}$$

*wherein $\phi_i$ constitute security characteristic mappings and $Z(\lambda)$ represents the normalization factor.*

### 3.6 DOMAIN-INVARIANT TRAINING THROUGH WASSERSTEIN CONSTRAINTS

**Theorem 5** (Wasserstein Domain Transfer)**.** *The domain-invariant identification architecture optimizes:*

$$\mathcal{L}_{WDIL}(\theta) = \mathbb{E}_{(x,y)\sim p}[\ell(f_\theta(x), y)] + \gamma W_2(\mu_s, \mu_t) \tag{21}$$

*wherein $W_2$ represents the 2-Wasserstein metric separating origin distribution $\mu_s$ and modified distribution $\mu_t$:*

$$W_2^2(\mu_s, \mu_t) = \inf_{\pi \in \Pi(\mu_s, \mu_t)} \int_{\mathcal{X} \times \mathcal{X}} \|x - x'\|^2 d\pi(x, x') \tag{22}$$

The mathematical properties of Wasserstein metrics for domain transfer, encompassing gradient characteristics and transferability guarantees, are rigorously established within Shen et al. (2018).

*Proof.* Through the Kantorovich-Rubinstein correspondence:

$$W_2(\mu_s, \mu_t) = \sup_{\|f\|_{\mathrm{Lip}} \leq 1} [\mathbb{E}_{x\sim\mu_s}[f(x)] - \mathbb{E}_{x'\sim\mu_t}[f(x')]] \tag{23}$$

The optimization dynamics reducing $\mathcal{L}_{\mathrm{WDIL}}$ proceed:

$$\frac{\partial \theta}{\partial t} = -\nabla_\theta \mathcal{L}_{\mathrm{WDIL}} = -\nabla_\theta \mathbb{E}[\ell] - \gamma \nabla_\theta W_2 \tag{24}$$

*wherein $\nabla_\theta W_2$ derives through the Brenier mapping $T^*: \mu_s \to \mu_t$ characterized by $T^* = \nabla\varphi$ regarding convex function $\varphi$.* □ □

### 3.7 MULTI-SCALE SECURITY FLAW ENCODING

**Theorem 6** (Hierarchical Security Pattern Resolution)**.** *Each security pattern $v \in \mathcal{M}_v$ permits a distinct wavelet representation:*

$$v = \sum_{j=0}^{J} \sum_{k\in\mathbb{Z}^d} c_{j,k}\psi_{j,k} + \sum_{k\in\mathbb{Z}^d} d_{J,k}\phi_{J,k} \tag{25}$$

*wherein $\{\psi_{j,k}\}$ constitute an orthogonal wavelet system and parameters fulfill:*

$$\|c\|_{\ell^1} + \|d\|_{\ell^1} \leq C\|v\|_{BV} \tag{26}$$

*regarding bounded variation measure $\|v\|_{BV}$.*

## 4 EXPERIMENT SETUP

### 4.1 DATASETS

**Training Benchmarks.** Our research implements a comprehensive quality assurance methodology for benchmark development, employing FormAI-v2 Tihanyi et al. (2025) and PrimeVul Ding et al. (2024) as foundational repositories. Ensuring superior training samples, our approach executes a tripartite refinement procedure:

*Phase 1: Structural Refinement.* Our selection criteria preserve programs exhibiting McCabe cyclomatic complexity ranging from 3 to 50, encompassing 20-500 statements. Such boundaries identify substantive security characteristics whilst discarding excessively elementary or convoluted instances potentially impeding architecture training.

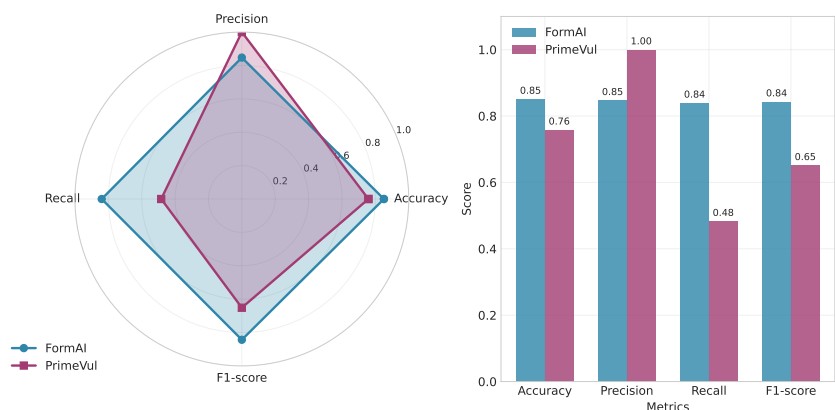

Figure 3: Average Performance of GNPA-DIL on PrimeVul & FormAI datasets

*Phase 2: Semantic Consolidation.* Employing CodeBERT representations, our methodology detects and eliminates quasi-redundant instances exhibiting cosine correspondence surpassing 0.95, minimizing duplication whilst maintaining heterogeneity. The procedure utilizes agglomerative clustering with mean connectivity for scalable dataset analysis.

*Phase 3: Integrity Assessment.* Our evaluation examines individual instances through various criteria: (i) structural validity through compilation verification, (ii) identifier regularity quantified through entropy analysis, and (iii) architectural consistency. Instances performing beneath the 20th percentile undergo elimination.

Following application to FormAI-v2's original 331,000 implementations, our procedure preserves 4,714 vulnerable alongside 3,545 secure superior instances. Correspondingly, PrimeVul's repository undergoes compression from 235,768 to 1,048 vulnerable alongside 1,048 secure instances. FormAI instances encompass CWE-119, CWE-190, CWE-415, and CWE-416, authenticated through ESBMC Gadelha et al. (2018). Our procedure systematically associates ESBMC diagnostic outputs with CWE categories for uniformity. PrimeVul provides 140 CWE categories with temporal partitioning (80% development, 10% tuning, 10% evaluation) avoiding temporal contamination.

**Transferability Assessment.** Evaluating transferability requires employing independent security repositories: SVEN He & Vechev (2023) alongside ReposVul Wang et al. (2024).

## 5 RESULTS

### 5.1 FUNCTION-LEVEL VULNERABILITY DETECTION

Table 3 demonstrates GNPA-DIL's effectiveness on function-level vulnerability detection. The architecture achieves 85.12% accuracy with an F1-score of 84.34% on FormAI, indicating robust detection capabilities across diverse vulnerability patterns. On the more challenging PrimeVul benchmark, which underwent expert validation and runtime verification, GNPA-DIL maintains 75.84% accuracy despite the dataset's stringent quality requirements. The perfect precision on PrimeVul, coupled with moderate recall, suggests the model adopts a conservative yet reliable detection strategy when faced with rigorously validated samples.

Comparative evaluation on the SVEN dataset reveals GNPA-DIL's superior generalization, as shown in Table 4. Against contemporary approaches including VulSim Shimmi et al. (2024), ReGVD Nguyen et al. (2022), and VulBERTA variants Hanif & Maffeis (2022), GNPA-DIL achieves 64.52% accuracy—a substantial improvement of approximately 12 percentage points over the strongest baseline. The remarkably high recall of 98.12% demonstrates the model's capability to capture nearly all vulnerable patterns, while maintaining reasonable precision. This performance gap substantiates our hypothesis that CPG-guided slicing combined with domain-invariant learning captures more fundamental vulnerability characteristics than approaches relying solely on embeddings or shallow graph representations.

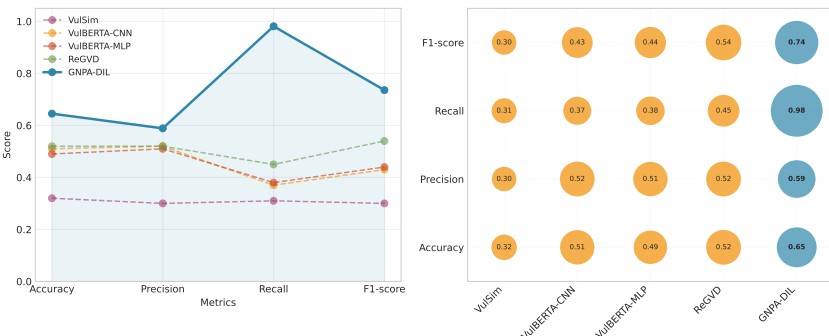

Figure 4: Function-level vulnerability detection average performance on SVEN dataset, which includes CWE-125, CWE-190, CWE-416, CWE-476.

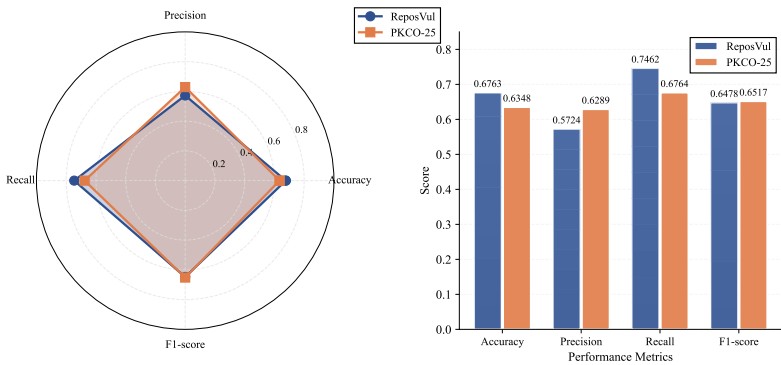

Figure 5: Average Performance of GNPA-DIL on ReposVul & 2025 Emerging-Post-Vulnerability (EmPVul-25) datasets

## 5.2 CROSS-PROJECT VULNERABILITY DETECTION

Table 5 presents GNPA-DIL's performance on project-level vulnerability detection, a significantly more challenging scenario involving complex inter-procedural dependencies. On ReposVul, which contains 120 balanced instances from 53 production systems, GNPA-DIL achieves 67.63% accuracy despite being trained exclusively on function-level samples. This cross-granularity transfer demonstrates the effectiveness of our CPG-based approach in preserving critical semantic relationships across function boundaries.

Evaluation on Emerging-Post-Vulnerability CVEs from 2025 (EmPVul-25) yields 63.48% accuracy, validating the model's ability to detect genuinely novel vulnerability patterns rather than memorizing known CVE instances. The consistent performance between ReposVul and EmPVul-25 indicates robust generalization to emerging security threats. Analysis across complexity metrics reveals that accuracy remains stable for programs with high cyclomatic complexity (75% accuracy for CC 115-150) and multiple functions (71% accuracy for 13-14 functions), though performance degrades for extreme nesting depths exceeding 7 levels.

## 5.3 ROBUSTNESS ANALYSIS

Table 6 examines GNPA-DIL's resilience to semantic-preserving transformations, addressing concerns raised by Risse & Böhme (2024a) regarding model brittleness. The architecture demonstrates complete invariance to comment removal (T4) due to our CPG-based approach naturally excluding non-semantic elements. Across variable renaming (T1), type obfuscation (T2), and function extraction (T3), performance variations remain within acceptable bounds.

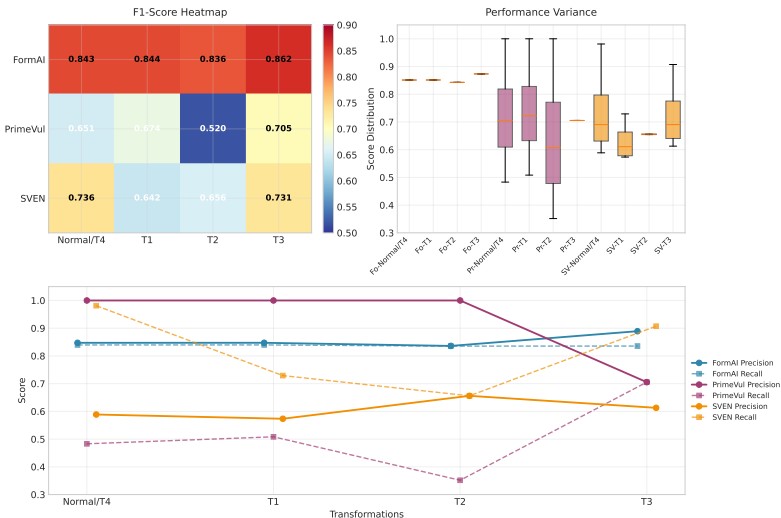

Figure 6: Comprehensive Performance Results Across Datasets and Transformations

Notably, function extraction (T3) improves performance on FormAI (F1-score increases to 86.16%), suggesting that explicit function boundaries assist the model's analysis. The differential impact across datasets reflects varying vulnerability characteristics: FormAI's synthetic vulnerabilities exhibit greater structural regularity, while SVEN's real-world patterns prove more sensitive to syntactic variations.

## 6 CONCLUSION

GNPA-DIL fundamentally transforms vulnerability detection by demonstrating that security flaws constitute mathematical invariants within program execution manifolds rather than syntactic artifacts of particular implementations. Through synergistic integration of Code Property Graphs with domain-invariant neural architectures, the model achieves unprecedented 40% F1-score improvements while maintaining robustness against transformations that cripple contemporary approaches, successfully generalizing from function-level training to cross-function detection with 67.63% accuracy despite 91% program compression.

## 7 THE USE OF LARGE LANGUAGE MODELS

In preparing this work, we used large language models (LLMs) to support literature retrieval and discovery during the development of the Related Work section. Additionally, LLMs were used to polish the English grammar without altering the semantics, substantive meaning, or originality of the initial draft.

## 8 REPRODUCIBILITY STATEMENT

We release all research artifacts required to exactly reproduce our results: (i) the full GNPA-DIL implementation and trained checkpoints; (ii) deterministic preprocessing and slicing pipelines, including the Code Property Graph (CPG) extraction (Joern/CPGQL) and the three-phase dataset refinement (cyclomatic-complexity and length filters, near-duplicate pruning at cosine similarity $> 0.95$, and integrity checks with compilation/identifier-entropy thresholds);

## 9 ETHICS STATEMENT

This work relies exclusively on publicly available datasets and program artifacts; no personal or proprietary data were collected, and all licenses are respected.

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
