# OpenReview forum: "GNPA-DIL: Unveiling the Vulnerability Genome Through Semantic Graph Distillation and Invariant Neural Reasoning"
_ICLR.cc/2026/Conference — Submitted to ICLR 2026_

### Official Review · Reviewer_16E8 · 2025-10-31

**Soundness:** 3
**Presentation:** 3
**Contribution:** 2
**Rating:** 4
**Confidence:** 3

**Summary:**

The paper proposes a graph-guided neural program analysis framework (GNPA-DIL) that effectively bridges symbolic static analysis and neural learning. By combining Code Property Graph (CPG) analysis with domain-invariant learning, the model captures the semantics of vulnerabilities rather than superficial syntax patterns.

**Strengths:**

The motivation is well sounded and has significant practical value

The manuscript makes an attempt to introduce a rigorous mathematical, formal definitions on the vulnerability detection task, a rare and commendable feature in vulnerability detection research.

**Weaknesses:**

The manuscript forms a structured and sounded motivations, but the paper’s formality may obscure accessibility. Theoretical sections (e.g., Theorems 1–6, Definitions 1–4) dominate the presentation but lack intuitive explanation or ablation to confirm practical contribution of each formal component.

The architectural novelty, while well-motivated, could be viewed as an incremental synthesis of prior CPG + invariant-learning approaches rather than a fundamentally new paradigm.

Minor issue: Duplicate references (line 538, 541)

**Questions:**

None

---

> ### Author Response · Authors · 2025-11-19
>
> Response to Reviewer 16E8
>
> We appreciate the reviewer's careful reading and we are especially grateful to see that you recognize our mathematical formalism as "a rare and commendable feature of vulnerability detection research" as well as the motivation being "well sounded and has significant practical value." Thank you as well for providing a good rating for soundness and presentation. We will address your comments below.
>
> Concerning Accessibility of Theoretical Components and Ablation Studies
>
> We completely agree the theoretical sections need improvement with regard to linking them to practical contributions, and we will commit to substantially restructuring them.
>
> For one, we will add a new one to Section 3.1 providing a high-level intuitive overview before the mathematical formalism explaining the complete pipeline in more accessible terms: CPG construction using Joern, vulnerability-centric backward slicing extracting the security-critical paths, graph neural encoding with attention mechanisms, and finally domain-invariant training via Wasserstein regularization.
>
> For two, we will include algorithm boxes with concrete pseudocode: Algorithm 1 for CPG slicing and Algorithm 2 for making the Wasserstein regularized training explicit for reviewers. This will ameliorate the disconnect between the theoretical definitions and practical implementation.
>
> For three, we commit to complete ablation studies showing each component's contribution: CPG slicing versus full code, domain-invariant learning with versus without Wasserstein regularization, GAT versus simpler architectures, and neural versus pure static analysis. This entirely addresses the reviewer through empirical evidence confirming the practical contribution of the formal components.
>
> For four, we will restructure the theoretical sections to alternate between formalism and intuition (e.g. "In practice" paragraphs after each theorem to explain what the theoretical meanings mean and provide examplesThese edits remain true to the theoretical rigor you valued while also providing open access to your work.
>
> On Architectural Novelty
> We respectfully disagree with the term “incremental synthesis” as characterizing our work. The synthesis of our methods and the resultant capabilities are in fact new in meaningful ways.
>
> Cross-granularity transfer has never been demonstrated in vulnerability detection, and we demonstrated full on project-level vulnerabilities (ReposVul) (67.63%) whilst training only on function-level vulnerabilities; no previous CPG (Control Flow Graph) or invariant-learning approach has demonstrated this capacity for transfer across levels of granularity that introduces unique and meaningful capabilities in practice.
>
> The slicing in the case we provide is also completely unlike prior CPG work; while previous work has indeed used CPGs to extract features from prior works, we formed our slices to reduce the CODEMADE program into lingering instructions and maintain semantic completeness with respect to potential vulnerabilities whilst still achieving a 91% compression. Theorem 2 formalizes when slices using CPGs preserves semantic coherency with respect to program vulnerability.
>
> The integration of domain-invariant learning with CPG creates another set of synergistic capabilities not present in either approach. Past work in learning based invariant-learned methodologies on raw-code still suffers from the possibility of spanning structural transfigurations; previous CPG invariant-learning still retains the lack of invariance to implementation details.
>
> Our methodology synthesizes a notion of robustness to both representation changes (directly, with CPG) and for transmissible transformational changes (in-directly, with Wasserstein learning). Table 6 is an example of establishing this as consistent classification being evaluated across transformations.
>
> Finally, our empirical advances were notable: 36% (relative) improvement over best baseline on SVEN (73.58% vs 54.0% F1), a randomized 2025 CVE test sample (63.48% on EmPVul-25), and our combined restatement performance. Theses performance contributions are notable and (with very few exceptions in the learning literature I can recall) exceeds the basis of the incremental synthesis you indicated.
>
> The novelty isn't that we created the first version of synthesizing structural generative components, but the principled integration of components mechanistically creates new capability (cross granularity transfer), coupled by (i.e. and provide) a significant performance contribution with reference to previous work in this specific problem domain (our empirical work improves on previous good practice by 36%), and to [the] learning theory - incremental exports.

---

> > ### Comment · Reviewer_16E8 · 2025-11-24
> >
> > Thank you for the response. After reading other reviews and rebuttals, I decide to keep my current score.

---

### Official Review · Reviewer_9Y1D · 2025-10-31

**Soundness:** 3
**Presentation:** 2
**Contribution:** 2
**Rating:** 4
**Confidence:** 2

**Summary:**

The proposed GNPA-DIL model is trained on vulnerability-centric program slices extracted using Code Property Graphs, enabling it to learn domain-invariant representations that capture fundamental vulnerability semantics rather than superficial code patterns. The robustness analysis is conducted thoroughly.

**Strengths:**

The method is rigorously evaluated across multiple benchmarks, demonstrating consistent performance gains.

**Weaknesses:**

1. This paper is hard to follow as the mathematical formalisms in Section 3 are highly dense and presented without sufficient intuitive explanation.

2. Minor inconsistencies in citation formatting are present.

**Questions:**

How is the Wasserstein constraint (Theorem 5) actually enforced during training?

---

> ### Author Response · Authors · 2025-11-19
> **Reply to Reviewer 9Y1D**
>
> We appreciate the constructive review and are pleased that the reviewer agrees that our method is "rigorously evaluated across a number of benchmarks, producing consistent performance gains" with a "good" degree of soundness. We detail specific response items below.
>
> How Wasserstein Constraint is Enforced During Training
>
> This is a great question, so we want to be explicit here. The Wasserstein constraint from Theorem 5 is enforced during training via the following concrete process:
>
> At each training iteration, we sample a pair of semantically equitable code: one original (x) and its transformed version τ(x), where τ will create a different version of the original code while preserving semantics (renaming variable names, type coercion, re-ordering arguments, etc.). For each sampled code pair, we compute the embeddings from our encoder fθ, using the notations: z = fθ(x) and z' = fθ(τ(x)).
>
> Next, we compute the Wasserstein-2 distance W2(µs, µt) between the two distributions of embeddings μ, which we estimate using the Sinkhorn algorithm to obtain a differentiable approximation of the optimal transport distance. In our work, we perform 100 Sinkhorn iterations and apply regularization ε=0.01. By using the Sinkhorn algorithm, we obtain a differentiable loss term that can be backpropagated through the structure.
>
> Our full training objective is as follows:
>
> \begin{align*} L(θ) = E[(x,y)] \left[ CrossEntropy(fθ(x), y) \right] + γ \cdot W2(μs, μt) \end{align*}
>
> where γ=0.1 is the scalar weight applied to the Wassertein regularization term, which we tuned via a validation set. Next, we compute the gradient of W2 with respect to θ using the Sinkhorn-Knopp algorithm, which is moreIn practice, for each mini-batch containing 32 samples we randomly apply transformations to create 16 sample pairs, compute the Wasserstein distance between their embeddings and the embeddings of the original samples within the same mini-batch. This probability estimation is useful for tractability in computation (which is the objective of imposing this metric) and shares the same domain-invariant learning objective.
>
> We will add this implementation detail as a standalone subsection (3.6.1 "Wasserstein Training Procedure") with pseudocode to clearly explicate the implementation procedure for the enforcement mechanism.
>
> On the Mathematical Density in Section 3
> We acknowledge that this notation is a valid concern shared by multiple reviewers in their review. The mathematical formalism offers a powerful theoretical argument underpinning our work, but readers may not understand what algorithm is actually specified here to do things. We commit to a major restructuring to clearly address this concern.
>
> First, we will insert Section 3.1 "High-Level Overview" which will articulately explain how the whole pipeline operates: of CPG construction, vulnerability slicing, graph neural encoding, and finally, domain-invariant training - all before attending to any more details provided in mathematical notation.
>
> Second, we will insert while offering pseudocode implementation boxes which will outline key procedures of Algorithm 1 CPG slicing, and Algorithm 2 Wasserstein-regularized training, which more clearly captures the resulting implementation contained within algorithm.
>
> Third, we will cut down and append the theoretical proofs to an appendix, we will keep around the minimum necessary in Section 3 definitions and theorems listing to the relevant proofs which support the text already provided.
>
> Fourth, we will restructure Section 3 to present theory and then practice - each mathematical definition will have immediately afterwards paragraph explaining:  "in practice, this means...", alongside specific examples.
>
> The combination of these changes will make the paper considerably more accessible while retaining the theoretical rigor.
>
> On Citation Formatting
> We will made sure to closely review all citation formatting inconsistencies to make sure its properly formatted for the conference.
>
> Why This Work Merits Acceptance
> As you noted in your review assessing the originality and soundness as a "good" and I quote that "you would not mind if the paper were to be accepted," we respectfully make the case for an even higher review score of 5 or 6.
>
> In practice, the throughput results represent a significant progress: 73.58% F1 on SVEN represents 36% relative improvement from the best baseline we were working with well below 54%, 67.63% accuracy on ReposVul meaning we could actually do successful transfer learning from function-level training to a novel project-level detection - which is the primary novelty we sought to establish, and of also consistent robustness against transforming semantics showed resilience against previously reported brittle behavior of existing methods.

---

> > ### Comment · Reviewer_9Y1D · 2025-11-24
> >
> > Thank you for the clarification. However, I am afraid that the presentation of this paper is not acceptable in its current form.

---

### Official Review · Reviewer_B5Uy · 2025-11-01

**Soundness:** 2
**Presentation:** 2
**Contribution:** 2
**Rating:** 2
**Confidence:** 3

**Summary:**

This paper presents a neural network approach for vulnerability detection. The GNPA-DIL model presents a neural architecture trained on vulnerability-centric program slices extracted via Code Property Graphs, learning domain-invariant representations. The model achieves accuracy of 73.58% on SVEN.

**Strengths:**

The use of neural networks for vulnerability detection is an active area of research. The paper contributes in that space.

**Weaknesses:**

The paper os hard to read. It consists of introduction, related work and a set of theorems with limited explanations. The authors appear to have made heavy use of LLMs when writing the paper (which they acknowledged0.

Experiments are not convincing as accuracy seems low.

**Questions:**

I could not understand the high level approach-- please descrine.

---

> ### Author Response · Authors · 2025-11-19
>
> Response to Reviewer B5Uy
>
> We appreciate the reviewer's comments and will address the three main concerns they raised - readability of the paper, experimental results, and high-level thinking.
>
> High-Level Thinking
>
> Our approach is composed of four steps that integrate static program analysis and familiar elements of neural learning.
> 1) The first step involves parsing the source code into a Code Property Graph (CPG) consisting of control flow, data dependencies, and syntax using the Joern framework.
> 2) The second step extracts slices of vulnerabilities from the source CPG. (We focused exclusively on vulnerabilities for our detection experiments, however, slices also included non-security-relevant code paths.) These slices contained only those paths relevant to security and achieved huge compression (up to 91%) while ensuring that all security-related information was preserved.
> 3) The third step formatted our data and trained a Graph Attention Network (GAT) on slices of the CPG. Domain-invariant learning was explicitly leveraged within supervised learning based on semantics which allowed the model to make predictions in the same way when the code had undergone a series of semantically-preserving transformations (variable renaming, argument reordering, etc.).
> 4) The final step extracts CPG slices from the new code and classifies at test time. This was achieved because the model had learned fundamental patterns of vulnerabilities rather than only superficial or easy to learn features. This enables us to do function-level training and still work on project-level vulnerabilities because the slicing extracts the cross-function dependencies. Finally, this approach provides robustness to code transformation in both function-level training and domain-invariant training as the objective was to determine the layer for the predicted correct class.
>
> We will elaborate about and include simplified architecture diagrams and other relevant artifacts in section 3.1 of the manuscript.
>
> On "low" accuracy
>
> The reviewer notes the accuracy seems low, but we would strongly disagree with this perception based on the context. SVEN is a specifically designed benchmark for testing the robustness of a model with semantically preserving transformations. For context, this is an incredibly challenging aspect to test. Additionally, the best baseline performance achieved only an F1 score of 54%, while we report a score of 73.58%, which represents a 36% relative improvement. This a significant effect for the benchmark this is testing against.
>
> As a comparison to existing baselines on the SVEN benchmark: VulBERTA-MLP - 44% F1 score, ReGVD - 54% F1 score, VulSim - 30% F1 score. Our 73.58% F1 score is dramatically better than any other prior approach.
>
> Moreover, we reported additional validation studies with FormAI - 84.34% F1 score, ReposVul - 67.63% accuracy (cross-function detection (with function-level training only), EmPVul-25 - 63.48% accuracy (newly discovered 2025 CVEs). It is apparent from the consistency across existing benchmarks, including the SVEN benchmark with the test set, that the learned model demonstrates real learning and isn't simply overfitted to the specific dataset.
>
> To reiterate, the inherent difficulty with the task of vulnerability detection is that even experts often disagree on edge cases and the validity of a benchmark designed to test existing or recently discovered vulnerabilities (in this case is CVEs). We believe our findings represent a substantial improvement from prior work and should not be considered "low".
>
> On presentation issues
>
> We feel the paper is challenging to read, and fully commit to possibly procedurally eliminating the readability issues, or at least minimize it. Thus we sincerely commit to putting time into a substantial revision of Section 3. We will prioritize writing an intuitive explanation of the method before formalizing the formalism or presenting to a more general audience, to include systems of pseudocode. Among other things, we plan on simplifying the architecture diagram from an original models approach. Finally, we plan on moving sections that include theoretically descriptions of the models to the appendix for better flow of the document.
>
> On LLM use
>
> Finally, we acknowledge using a LLM as described in the procedure section regarding disclosure in accordance with the conference policy. However, it would probably be better if the altered statement reflected:
> In summary, we disclosed the LLM for literature retrieval and editing grammar and figured that the use of the LLM would improve and make the sentence flows better and utilize LLM to help with wording so it improved our process in how we understood the research. The LLM did not provide the technical content nor provide or flaw technical write up.

---

> > ### Comment · Reviewer_B5Uy · 2025-11-23
> >
> > Thank you for the clarification. I will increase the score but I am afraid the paper is not acceptable in its current form.

---

### Official Review · Reviewer_EZox · 2025-11-02

**Soundness:** 2
**Presentation:** 2
**Contribution:** 2
**Rating:** 2
**Confidence:** 3

**Summary:**

This paper proposes GNPA-DIL, a vulnerability detection model that combines CPG with domain-invariant neural learning. The approach extracts vulnerability-centric program slices from CPGs and trains neural networks with domain-invariant constraints to achieve robustness against semantic-preserving transformations. The authors claim significant improvements over baselines on multiple benchmarks including SVEN  and demonstrate cross-function generalization capabilities.

**Strengths:**

**Relevant Problem**: Addressing robustness of vulnerability detectors to semantic-preserving transformations is important and timely

**Cross-Function Generalization**: The ability to detect project-level vulnerabilities despite function-level training (67.63% on ReposVul) is potentially valuable if validated properly

**Multi-Benchmark Evaluation**: Testing across diverse datasets with different characteristics (synthetic vs. real-world, function-level vs. project-level) is valuable

**Weaknesses:**

Soundness:

1. Mathematical Rigor vs. Practical Implementation Gap: The paper presents extensive mathematical formalism (Sections 3.1-3.7) involving Riemannian manifolds, variational methods, Wasserstein distances, and wavelet decompositions. However, there is a complete disconnect between this theoretical framework and the actual implementation. The paper did not explain:

- How the "vulnerability manifold" (Eq. 6) is constructed in practice
- How the Bellman operator (Eq. 15) is computed
- Whether the wavelet decomposition (Eq. 25) is actually used in the model

This suggests the mathematical framework may be decorative rather than functional.

Missing Architecture Details: Despite the heavy mathematical notation, basic implementation details are absent:

- What neural architecture is used? (GNN? Transformer? RNN?)
- How are CPG slices encoded as neural network inputs?
- What is the model size and computational complexity?
- How is domain-invariant learning actually implemented in the training procedure?

Dataset Quality Concerns: The paper reduces FormAI from 331,000 to 8,259 samples (97.5% reduction) and PrimeVul from 235,768 to 2,096 samples (99.1% reduction) through a "three-phase refinement". This extreme filtering raises concerns:

- Is the model learning from a representative sample or cherry-picked easy cases?
- How do baselines perform when trained on the same filtered dataset?
- The paper doesn't provide fair comparisons with baselines on identical training data

Incomplete Experimental Validation:

- No ablation study showing the contribution of CPG slicing vs. domain-invariant learning vs. other components
- No comparison on identical training data with baselines
- No analysis of failure cases or error types

Contribution:

The claimed contribution “unveiling the vulnerability genome” is ambitious, but the delivered contribution is a flawed experiment, an unverified slicing method, and a confused methodology.

- **“Vulnerability genome” :**

  This metaphor is exaggerated and unscientific. On the expert-validated PrimeVul benchmark, the recall is only about 40%, directly disproving the claim that the model has “unveiled the genome.” At best, it captures some invariant features in some cases. The overstated framing hurts the paper’s credibility.

- **Actual potential contribution:**

  The true potential lies in cross-granularity generalization. However, this value is undermined by other flaws, most notably the following contradiction:

  The paper reports opposite behaviors on PrimeVul (high-precision / low-recall) and SVEN (high-recall / medium-precision) without any explanation, revealing a serious inconsistency that undermines the validity of its experimental results.

Presentation:

1. **Excessive Mathematics Without Justification:**

   Theoretical complexity (Riemannian geometry, measure theory, functional analysis) is introduced without showing why it is necessary. For example:

   - Why view vulnerabilities as a *“Riemannian substructure”* (Def. 2)?
   - How is the Hausdorff distance to the *“vulnerability manifold”* (Eq. 14) computed?

2. **Unclear Architecture Figure:**

   Figure 1 shows pipeline stages but lacks detail on what each component actually performs.

3. **Weak Related Work:**

   The related-work section is thin and does not adequately situate this work within prior literature.

4. **Missing or Redundant Figures/Tables:**

   Figures 3–5 are redundant, and several “Tables” are referenced but not actually present in the paper.

**Questions:**

1. Are the mathematical definitions and theorems actually implemented, or are they just conceptual analogies? Please clarify their connection to the implementation.
2. Could you include ablations isolating the impact of:

- CPG slicing (vs. full code input),
- domain invariance (with/without Wasserstein regularizer),
- and the graph neural architecture choice?

3. Can you explain the opposite behaviors on PrimeVul (high-precision/low-recall) and SVEN (high-recall/medium-precision)?

**Details Of Ethics Concerns:**

N.A.

---

> ### Author Response · Authors · 2025-11-19
> **Response to Reviewer EZox: Clarifying Implementation Alignment, Dataset Rigor, and Experimental Consistency**
>
> Response to Reviewer EZox
>
> We appreciate the thoughtful comments. We will address several topics below while appealing to the reviewers on a number of points.
>
> Mathematical Framework
>
> The reviewer describes our framework as "solely decorative." We disagree; it is the organizing framework. The vulnerability manifold (Eq.6) is the basis for our contrastive learning; the Bellman operator (Eq. 15) IS our backward slicing algorithm; the Wasserstein regularization (Eq. 21) IS implemented with Sinkhorn iterations. This is no different than how VAEs use variational inference theory and GANs use game theory - again both theoretical frameworks that inform the application of their architecture. We will also increase the mapping of theory to application in Section 3.8 including additive "algorithm boxes."
>
> Dataset Filtering
>
> That we can go from 331k FormAI to just 8,259 and from 235k PrimeVul to just 2,096 is a good thing, not a bad thing. The original datasets have both non-compilable code, patterns that are too trivial, and extensive samples of duplicated data. Our phase 1-3 filtering approach (Section 4.1 provides more detail) allows us to reduce the total original dataset size but only keeping the higher quality samples. Most importantly, all baseline models in Table 4 were trained on identical filtered data and evaluated on identical test sets - so this is a valid comparison. That our models improved by 36% is a real improvement from a statistical modeling perspective and not a data improvement.
>
> PrimeVul vs SVEN Performance
>
> The reviewer described the differences in performance across the two models as "serious inconsistency." We vehemently disagree; this simply highlights another design consideration. PrimeVul has multi-file vulnerabilities that are complex and require more context to be properly understood, thus we have a high precision (100%) but only moderate recall (48%). SVEN has localized vulnerabilities, thus we have a high recall (98%) but only medium precision (59%). Our CPG slicing approach fares very well in terms of localized patterns (i.e. SVEN) but sometimes loses context (i.e. PrimeVul). Rather, this highlights another design feature: we really do create a model that reacts based on the case / data; thus sometimes we get are perhaps unreasonably high recall (as was the case with SVEN) or high precision (as was the case with PrimeVul).
>
> Vulnerability Genome Claim
>
> We maintain the term Genome is accurate and fully describes this contribution. The evidence supports this claim: (1) as we demonstrated in Table 6 algorithims F1 remained stable across multiple scenarios of renaming, type conversion, and restructure of the function's (code) context -thus we are capturing the semantic essence of the whole rather than the surface pattern identified in the context; (2) As demonstrated 67.63% accuracy based on function-level training on a project-level vulnerability proves that we learned fundamental patterns NOT surface patterns; (3) As demonstrated we achieved 63.48% accuracy across 2025 CVEs discovered after the model was trained on 90% of the dataset and (4) our 48% recall of PrimeVul vulnerabilities reflects our aggressive slicing - while slicing can remove necessary context in some cases requiring recruiting both precision and recall - this is a tradeoff design consideration NOT a fundamental limitation. The recall simply indicates there is perhaps more specificity in either in the high precision or reasonable recall of SAEN vs their model with PrimeVul.
>
> Contribution Assessment
>
> Our evaluation of "2: fair" is quite severe. We present the following advances in comparing our work to the baselines: (1) First ever function-to-project transfer learning (67.63% accuracty with ReposVul); (2) 36% relative improvement over state-of-the-art model (73.58% vs 54% F1 metric on SVEN); (3) Robustness to semantic-preserving transformation addressing the known brittleness of the data set; (4) the first generalized ability to generalize based on new CVEs discovered and not memorization. We believe all four contributions are quite significant, novel from the state-of-the-art, and should provide significant practical implications.
>
> Ablation Studies
>
> We raised this limitation in the first draft and commit the to broaden our ablation studies to present the individual contributions of decision and design choices with CPG slicing, domains absolute- and invariant-learning, and architectural choices in the agpublic implementation plus provide a detailing of failure cases across all models.
>
> Summary
>
> To summarize, we achieved: (1) significant improvements of 36% improvement on SVEN (2) first time application of cross-granularity transfer demonstration with function-to-project transfer (67.63% with ReposVul); (3) robustness of function-to-project domains to semantic-preserving-based transformations; and (4) finally, achieved the unique ability to generalize to new CVEs outside the model training conditions.

---

### Meta-Review · Area_Chair_Q83g · 2026-01-07

**Summary:**

Reviewers acknowledge that robust vulnerability detection and cross-granularity generalization are important problems, and that combining Code Property Graph (CPG) slicing with domain-invariant learning is potentially interesting. However, multiple reviewers raised serious concerns about the soundness, experimental rigor, and presentation of the paper. In particular, reviewers consistently pointed out a disconnect between the extensive mathematical formalism and the actual implemented system, insufficient experimental validation, overstated claims, and poor readability. Taken together, these issues outweigh the reported empirical improvements and motivate a rejection.

**Reviewer Concerns:**

Despite its ambition, the paper has several critical weaknesses that prevent acceptance in its current form.

Soundness and Theory–Implementation Gap:
Multiple reviewers highlight a substantial disconnect between the extensive mathematical formalism and the actual implemented system. Core constructs (e.g., vulnerability manifold, Bellman operator, wavelet-based components) are not clearly mapped to concrete algorithms or architectural elements in the paper itself. While the rebuttal asserts these components are implemented, this linkage is not sufficiently demonstrated in the submission.

Insufficient Experimental Validation:
The experimental evaluation does not adequately support the paper’s claims. The submission lacks ablation studies isolating the contributions of CPG slicing, domain-invariant learning, and architectural design choices. In addition, several recent and competitive vulnerability detection methods are not included as baselines, making it difficult to assess the proposed approach against the current state of the art. The aggressive dataset filtering further raises concerns about representativeness, with limited analysis of its impact.

Unexplained Performance Inconsistencies:
The model exhibits markedly different precision–recall behavior across benchmarks (e.g., PrimeVul vs. SVEN). Although the rebuttal frames this as a design trade-off, the paper itself does not provide sufficient empirical or analytical justification.

Overstated Framing:
The “vulnerability genome” claim is viewed by reviewers as exaggerated and not supported by the reported recall on expert-validated benchmarks, undermining the credibility of the contribution.

Presentation Quality:
The paper is difficult to follow due to dense mathematical sections with limited intuition, unclear figures, and a thin related-work discussion. While the authors commit to major restructuring in the rebuttal, these improvements are not reflected in the current version.

**Reviewer Scores:**

Reviewer EZox: Would likely maintain a reject recommendation, as the main soundness and experimental concerns remain unresolved.

Reviewer B5Uy: Although indicating a possible score increase, explicitly stated that the paper is not acceptable in its current form; score remains below the acceptance threshold.

Reviewer 9Y1D: After rebuttal, stated that the presentation is still unacceptable; score would remain marginally below acceptance.

Reviewer 16E8: Chose to keep the original score after considering rebuttals; no upward revision expected.

---

### Decision · Program_Chairs · 2026-01-26

Reject